# Frequency and Spectrum of Mutations Induced by Gamma Rays Revealed by Phenotype Screening and Whole-Genome Re-Sequencing in *Arabidopsis thaliana*

**DOI:** 10.3390/ijms23020654

**Published:** 2022-01-07

**Authors:** Yan Du, Zhuo Feng, Jie Wang, Wenjie Jin, Zhuanzi Wang, Tao Guo, Yuze Chen, Hui Feng, Lixia Yu, Wenjian Li, Libin Zhou

**Affiliations:** 1Biophysics Group, Biomedical Center, Institute of Modern Physics, Chinese Academy of Sciences, Lanzhou 730000, China; duyan@impcas.ac.cn (Y.D.); fengzhuo@impcas.ac.cn (Z.F.); kirawang@impcas.ac.cn (J.W.); jinwenjie@impcas.ac.cn (W.J.); wangzz@impcas.ac.cn (Z.W.); cyz111abc@163.com (Y.C.); fengh1992117@163.com (H.F.); yulx@impcas.ac.cn (L.Y.); wjli@impcas.ac.cn (W.L.); 2University of Chinese Academy of Sciences, Beijing 100045, China; 3National Engineering Research Center of Plant Space Breeding, South China Agricultural University, Guangzhou 510640, China; guoguot@scau.edu.cn; 4Kejin Innovation Institute of Heavy Ion Beam Biological Industry, Baiyin 730900, China

**Keywords:** gamma rays, mutation, phenotype screening, whole-genome re-sequencing, *Arabidopsis thaliana*

## Abstract

Genetic variations are an important source of germplasm diversity, as it provides an allele resource that contributes to the development of new traits for plant breeding. Gamma rays have been widely used as a physical agent for mutation creation in plants, and their mutagenic effect has attracted extensive attention. However, few studies are available on the comprehensive mutation profile at both the large-scale phenotype mutation screening and whole-genome mutation scanning. In this study, biological effects on M_1_ generation, large-scale phenotype screening in M_2_ generation, as well as whole-genome re-sequencing of seven M_3_ phenotype-visible lines were carried out to comprehensively evaluate the mutagenic effects of gamma rays on *Arabidopsis thaliana*. A total of 417 plants with visible mutated phenotypes were isolated from 20,502 M_2_ plants, and the phenotypic mutation frequency of gamma rays was 2.03% in *Arabidopsis thaliana*. On average, there were 21.57 single-base substitutions (SBSs) and 11.57 small insertions and deletions (InDels) in each line. Single-base InDels accounts for 66.7% of the small InDels. The genomic mutation frequency was 2.78 × 10^−10^/bp/Gy. The ratio of transition/transversion was 1.60, and 64.28% of the C > T events exhibited the pyrimidine dinucleotide sequence; 69.14% of the small InDels were located in the sequence with 1 to 4 bp terminal microhomology that was used for DNA end rejoining, while SBSs were less dependent on terminal microhomology. Nine genes, on average, were predicted to suffer from functional alteration in each re-sequenced line. This indicated that a suitable mutation gene density was an advantage of gamma rays when trying to improve elite materials for one certain or a few traits. These results will aid the full understanding of the mutagenic effects and mechanisms of gamma rays and provide a basis for suitable mutagen selection and parameter design, which can further facilitate the development of more controlled mutagenesis methods for plant mutation breeding.

## 1. Introduction

Continuous improvement of plants is essential to alleviate the increasing pressure on food security and the diversification of daily life demands, including population growth, declining crop production due to climate change, changing food preferences, and increasing dependence on ornamental plants. Genetic variation is an important source of plant improvement and adaptation, as it provides various alleles that contribute to the development of new traits for plant breeding. The application of mutation techniques, including physical mutagens (e.g., gamma rays, X-rays, heavy ions, and protons), chemical agents (e.g., ethyl methanesulfonate (EMS), N-nitroso-N-methylurea, and colchicine), and bio-techniques (e.g., genetically modification, transgenic or gene editing based on the CRISPR/Cas9 technology), has created a large number of genetic variations, which plays a crucial role in plant breeding, genetics, and advanced genomics studies since 1928. More than 3365 new varieties in hundreds of plant species produced by mutagenesis were officially registered according to the FAO/IAEA mutant variety database, bringing billions of dollars in additional revenue (until 15 November 2021, https://mvd.iaea.org/#!Home). Among these new varieties, 2610 were derived from physical mutagens; furthermore, up to 65.25% (1703) were induced by gamma-ray irradiation. In addition to being directly released as varieties, the induced mutants can also be used as parents in hybridization programs for plant improvement. For instance, a high-yielding, low-phytate basmati rice cultivar was developed by combining hybridization, backcross, and maker-assisted breeding based on the low-phytate mutants Lpa5, Lpa9, and Lpa59 that induced by gamma rays (^60^Co, 250 Gy) [1].

Gamma rays belong to photon radiation, which can induce multiple types of DNA damage, ranging from nucleotide modifications to DNA strand breaks (e.g., oxidized base, abasic sites, single-strand breaks (SSBs), double-strand breaks (DSBs)). If this DNA damage fails to be repaired or is repaired imprecisely, mutations such as single-base substitutions (SBSs), deletions, insertions, inversions, or translocations, may occur at the genome scale and finally lead to changes in the phenotypic traits [2,3,4,5,6,7]. For instance, in an *Arabidopsis* transgenic line containing the *Escherichia coli rpsL* gene, the frequency of SBSs was significantly increased by gamma rays [8]. An SBS was induced by gamma rays in the second exon of *sd-1*, which encodes the gibberellin 20-oxidase, and this mutation led to a semidwarf and high potential yield phenotype [9]. Gamma rays lead to a distinct genome mutation profile in higher plants when compared to EMS, which mainly causes nucleotide modifications that induce C to T mispairing, resulting in the transition of C/G to T/A [10,11]. This difference is a consequence of repairing ionizing-induced DNA damage, especially the DSBs. In higher eukaryotes, the classical non-homologous end joining (C-NHEJ) is the predominant repair pathway for the majority of DSBs. C-NHEJ has been considered to be error prone and results in small InDels around the rejoint sites [3].

The mutagenic effects and mutation characteristics of gamma rays in plants have been particularly important issues concerned by geneticists and breeders. The importance of understanding the relationship between the mutagenesis parameters (such as physical parameters, tissues, sample status, and so on) and the mutation effect is increasingly being recognized [12,13,14,15,16,17,18,19,20,21,22,23,24,25,26,27,28]. The main parameters of gamma rays are the total dose and dose rate. Their effects have been investigated by determining multiple biological endpoints. For instance, in *chrysanthemum morifolium* cv. ‘Taihei’ (Yamaguchi), high total doses at low dose rates of gamma-ray irradiations could efficiently induce mutations (flower color mutation frequency with less radiation damage (nuclear DNA content)). By analyzing the radiation sensitivity, mutation frequency, and spectrum of mutants induced by different parameters of gamma rays, Kim found that the optimal conditions for tissue culture of *Cymbidium* hybrid cultivar RB003 were 35 Gy/4 h (mutation frequency, 4.06%), for RB012 was 20 Gy/1 h (mutation frequency, 1.51%) (Kim, 2020). In addition, the sample itself, including status, genotypes are also another important aspect that affects the mutagenesis. The optimal dose of gamma rays for mutation induction in ginseng (*Panax ginseng*) was tissue dependent; for instance, <20 Gy for one-year-old roots, 40 Gy for dehiscent seeds, and 60–80 Gy for somatic embryos [24]. Comparing the agronomic traits of 30 M_2_ lines and the corresponding M_3_ lines derived from gamma rays in two Canola (*Brassica napus* L.) cultivars, Emrani showed that the response to gamma rays was genotype dependent in terms of induction of genetic variations. Based on lethal dose (LD)_30_, LD_50,_ and reduction dose (RD)_50_ of 20 plant species that irradiated by gamma rays, Kim found that the radiation sensitivity was dependent on the plant families and genera, as well as the size of the genome [27].

The genome mutation profile is another important index for evaluating the mutagenic effects of gamma rays. In the last 10 years, the availability of next-generation sequencing (NGS) technology has greatly propelled the understanding of the genome-wide characteristics of mutations induced by gamma rays [10,29,30,31,32,33,34,35,36,37,38]. The first attempt was made by Henry. The insertions and deletions (InDels) mutations were detected in approximately 500 F_1_ seedlings produced by pollinating *Populus deltoides* with gamma rays-irradiated *Populus nigra* pollen based on HiSeq2000 and HiSeq2500 platform [30]. Very recently, using whole-genome sequencing analysis, they have demonstrated that gamma rays can trigger chromoanagenesis in poplar plants [29]. Datta established a low-coverage whole-genome sequencing approach to recover the large genomic InDels (ranging from 0.3 to 3.8 Mb) and copy number variation caused by gamma rays in triploid bananas [39]. In addition, the remaining studies are mainly focused on the features and frequencies of SBSs and small InDels by re-sequencing analysis of the lines in M_2_ generation or later. For instance, gamma rays induced a higher percentage of InDels in the genome of the tomato genome when compared to EMS [10]; it induced fewer InDels and complex SVs but more small mutations than carbon ions in rice mutants (M_4_–M_7_) and in randomly selected M_2_
*Arabidopsis thaliana* lines [31,35,40]. The genome mutation profile was also impacted by sample status, physical parameters, and so on. For instance, in pepper (*Capsicum annuum* L.), compared to seeds treatment, exposure of female or male gametophytes to the gamma rays led to similar or higher mutation frequency and spectrum in the progeny based on the genotype-by-sequencing analysis [38]. Hase also found that the sample status (seeds and seedlings), as well as the dose rate (acute and chronic radiation), did affect the genome mutation profile in *Arabidopsis thaliana* [31,41].

Many mutants have been isolated from progenies that derived from the seeds irradiated by gamma rays, including rice, *Arabidopsis thaliana*, sorghum, *NicotianaN. tabacum*, *Lathyrus sativus* and so on [1,8,16,19,42]. However, in most cases, it mainly focused on chlorophyll, dwarf, flower, and seed colors, and few mutant collections were aimed at multiple mutation traits based on large-scale mutation screening. In addition, many genetic modifications and gene editing crops have also been developed to improve the traits of interest [43]. However, mutation isolation that is limited to the desirable agricultural traits and pays little attention to the genetic resource collection for a long time may lead to genetic bottlenecks [44,45,46]. Furthermore, in the available studies, the phenotype mutation profile and mutant collections were always separated from the research of genome-scale mutation. To provide useful guides for researchers with different demands, in this study, we assessed the mutagenic effects of gamma rays on *A. thaliana* at both a phenotype and genome scale: 1) large-scale mutation screening for multiple category phenotypes and mutant collection; 2) characterization of genomic mutations in phenotype-visible populations using whole-genome re-sequencing, including the frequency, distribution, nucleotide bias, genes affected and NHEJ traces in the sequence context of SBSs and small InDels.

## 2. Results

### 2.1. Biological Effects of Gamma Rays on M_1_ Plants

The survival rate, elongation of the primary root and hypocotyl, and genetic polymorphisms [12] in the M_1_ generation were determined (Figure 1). Overall, radiation exposure caused detectable changes in these aspects. The growth of primary root was more sensitive to gamma rays than survival rate and the growth of hypocotyl. In detail, the primary root length was statistically different from that of the control when the dose was up to 200 Gy, and it was 71.81% that of control. When the dose was increased to above 1200 Gy, the growth of the primary root was completely inhibited. The survival rate of M_1_ plants was comparable with that of the control group at doses below 600 Gy (88.49% of that of the control) and showed a significant decrease at doses above 800 Gy. The length of the hypocotyl was significantly shorter than that of the control when the dose was higher than 500 Gy. The degree to which the skotomorphogenesis of *Arabidopsis thaliana* was impacted seemed to be lower than that of the primary root growth and survival rate since, at the highest dose, the hypocotyl length, primary root length, and survival rate were 55.00%, 5.84%, and 13.998% of the control, respectively. The genetic polymorphisms were detected in M_1_ populations that derived from 1000 Gy irradiated group (with a relative survival rate, primary root, and hypocotyl length of 62.17%, 21.87%, and 61.50%, respectively) by using inter-simple sequence repeat (ISSR) and random amplified polymorphic DNA (RAPD) techniques [12]. The result indicated that the effectiveness of gamma rays on genetic material. In the subsequent study, we chose the 1000 Gy irradiated group to perform large-scale mutation screening.

### 2.2. Spectrum and Frequency of Phenotype Mutations Identified in M_2_ Generation

A total of 20,502 M_2_ plants derived from 1139 M_2_ families were used for mutation screening. Regarding macroscopic morphology, a total of 417 plants from 157 M_2_ families were identified as morphological mutants (Table 1). The total percentage of phenotypic mutation was 2.03% in the M_2_ generation. Leaf morphological mutations (1.30%) were the most frequent type, followed by premature mutations (0.37%), seeds viability (0.18%), stem mutation (0.14%) and fertility mutation (0.05%). Leaf mutations consisted of five sub-groups, including mutations referred to lamina (50.19, 134/267), color (23.60%, 63/267), petiole (16.85%, 45/267), vein (4.87%, 13/267), and arrangement (4.49%, 12/267). Figure 2 shows some of the mutants isolated in the present study. All the results indicated that gamma rays were effective mutagen for the enrichment of phenotypic diversity in plants, especially for creating leaf mutations.

### 2.3. Type and Frequency of Genomic Mutations Induced by Gamma Rays in M_3_ Generation

Seven M_3_ individuals with visible mutant phenotypes, which including G200 (Figure 2b), G240 (Figure 2c), G266 (Figure 2y), G287 (Figure 2d), G320 (Figure 2e), G431 (Figure 2f) and G692 (Figure 2g), were re-sequenced. The bioinformatics pipeline detected 232 variants (Appendix A), which consisted of 151 (65.09%) SBSs, 38 (16.37%) single-base deletions (−1), 16 (6.90%) single-base insertions (+1), 19 (8.19%) deletions with a size ≥2 bp (≥2 bp Dels) and eight insertions (3.45%) with a size ≥2 bp (“≥2 bp Dels” and “≥2 bp Ins”) (Figure 3A). Among these mutations, 98 were heterozygous and 134 were homozygous. On average, 33.14 mutations were detected in the individual M_3_ line (Figure 3B). The total mutation frequency was 2.78 × 10^−7^/bp/plant, following the order: “SBSs” (1.81 × 10^−7^/bp) > “−1” (0.46 × 10^−7^/bp) > “≥2 bp Dels” (0.23 × 10^−7^/bp) = “+1” (0.19 × 10^−7^/bp) = “≥2 bp Ins” (0.10 × 10^−7^/bp) (Figure 3C).

### 2.4. Distribution of Mutations on Genome

To investigate whether there is a preference for the chromosome, reference sequence at which the mutation occurred, and the gene structure, the distribution of the genome mutations were conducted. All the 232 small variations were mapped to chromosomal coordinates. The data indicated that mutations were evenly distributed on each chromosome. There was no statistical difference in mutation frequency among five chromosomes (Figure 3D). By analysis of the reference sequence of the position at which the mutation occurred, no bias to the nucleotide was observed in each mutation type (Figure 4A). The distribution of mutations across the genome and its mutation effects on gene function were predicted using SnpEff and verified by Integrative Genomics Viewer (IGV). For SBSs, most of the mutations were located in the upstream and downstream intergenic regions, especially in the upstream, followed by the exon, and then untranslated region (UTR), splice region, intron, transposable, intergenic (Figure 4B). While for small InDels, more mutations were located in upstream and downstream intergenic regions, mutation numbers in other regions were at the same level (Figure 4B).

### 2.5. Characteristics of the SBSs and Small InDels Induced by Gamma Rays

The nature of the induced mutations, including the form and frequency of SBSs and the size distribution of small InDels, is useful to represent the characteristics of a physical mutagen. The average ratio of transitions (Ts) to transversions (Tv) was 1.60 in M_3_ plants. The distribution proportion of SBSs followed in order: “G/C>A/T” > “A/T>G/C” ≥ “A/T>T/A” ≥ “G/C>T/A” ≥ “A/T>C/G” = “C/G>G/C” (Figure 5A). The size of the small InDels detected ranged from one bp to 10 bp, and the single-base InDels were the most predominant type of InDels (Figure 5B). More in detail, the single-base deletion of adenine was the most prevalent type of InDels, followed by the single-base deletion of thymine (Figure 5B). The sequence context of small InDels was examined to check the terminal microhomology (homopolymeric sequence and polynucleotide repeats) at their rejoined sites (Figure 5C). For small InDels, 69.14% (56 out of 81 events) of the mutations were accompanied by terminal microhomology at the rejoined sites. Among these, 30 events involved with polynucleotide repeats and 26 events with homopolymeric. Among the 54 single-base InDels, eight events (14.81%) were involved with terminal polynucleotide repeats, while 26 events (48.15%) were involved with homopolymeric sequence. Regarding the InDels with a size ≥2 bp, 22 out of 27 events (81.48%) referred to the terminal microhomology, and the proportion of polynucleotide repeats was increased to 66.67% (18 out of 27 events). The sequence context of SBSs was also investigated (Figure 5C). Eighteen out of 28 (64.29%) C >T events exhibited a pyrimidine dinucleotide sequence. Meanwhile, we found that 33.11% (50 out of 151 events) of the SBSs occurred in the presence of homopolymeric sequences (25 events) and polynucleotide repeats (25 events). The flanking sequences around the non-G/C>A/T (44.68%, 42/94) mutations exhibited a higher proportion of terminal microhomology than those of the G/C>A/T mutations (14.03%, 8/57). The sizes of the terminal microhomology were also examined. For InDels with a size ≥2 bp, the terminal microhomology used ranged from 1 to 4 bp. The most commonly used microhomology size was 1 bp for single-base InDels and SBSs, while a longer terminal microhomology size, usually 2 bp, was applied for ≥2 bp InDels (Figure 5D, Appendix A).

### 2.6. Gene Affected by Gamma Rays

The 232 genomic variants detected by using whole-genome re-sequencing were located in 227 genes (Appendix A). The variants that were located in the exon but resulted in synonymous mutations, intergenic and intron regions were ruled out, and 63 genes remained for further analysis. On average, nine genes were predicted to be affected in each M_3_ line (Figure 6A and Table 2). The gene mutation frequency induced by gamma rays was 3.28 × 10^−4^ (9/27,411, a total of 27,411 protein-coding genes in *Arabidopsis thaliana* reference genome). To further investigate the functions and pathways of the affected genes involved, all 63 genes were subjected to Kyoto Encyclopedia of Genes and Genomes (KEGG) pathway and Gene Ontology (GO) analysis (Figure 6B). KEGG pathway analysis revealed that the affected genes were involved in 18 pathways, including the mRNA surveillance pathway, photosynthesis antenna proteins, carotenoid biosynthesis, metabolic pathways, ubiquinone and other terpenoid-quinone biosynthesis, nitrogen metabolism, pyrimidine metabolism, cyanoamino acid metabolism, photosynthesis peroxisome, pentose and glucuronate interconversions, purine metabolism, ubiquitin-mediated proteolysis, oxidative phosphorylation, starch and sucrose metabolism, phenylpropanoid biosynthesis, plant-pathogen interaction, and the biosynthesis of secondary metabolites. GO annotation analysis showed that the 63 affected genes were enriched in 39 biological process terms, 60 cellular component terms, and 46 molecular function terms. The detailed GO terms are listed in Appendix A.

## 3. Discussion

An important step in any project that uses the induced mutations for breeding and functional genome research is the establishment of a suitable, optimized, and economical strategy for mutagenesis. Therefore, it is of great importance to investigate the biological effect in M_1_ and the mutation profile in the later generations to guide mutagen selection and parameter optimization [27,33]. This information is especially significant for plants that are large and require huge spaces, as well as plants that not only have a long growth cycle but whose culture is also labor-intensive [39].

### 3.1. Biological and Phenotypic Effects of Gamma Rays Irradiation on Arabidopsis thaliana

To find a suitable dosage for large-scale mutation screening, the post-embryo development in M_1_ first, such as the survival rate, primary root, and hypocotyl growth, was investigated. The impact of gamma rays on photomorphogenesis was greater than that of skotomorphogenesis since the primary root development under light conditions was much more sensitive to gamma rays than the survival rate and hypocotyl length (Figure 1). A similar tendency was observed for the effects of carbon ions and gamma rays on the post-germination of *Arabidopsis thaliana* and *Triticum monococcum* L. cultivar Einkorn, respectively [23,47]. It is well known that the development of an embryo involves cell expansion, proliferation, and division in the root apical meristem (RAM) and shoot apical meristems [48]. Unlike hypocotyl elongation that is the result of cell expansion and elongation, the root is mainly initiated from de novo cell division; therefore, it is more sensitive to mutagenesis [49]. In addition, based on the organic development sequence, root emergence occurred earlier than that of other organs, and the chance of complete damage repair was relatively lower than that observed in later development.

The mutation frequency and spectrum of the phenotypes induced by ionizing radiation differ among plant species, sample status, the homozygosity or heterozygosity of the starting materials. In the present study, a total of 417 plants with visible mutated phenotypes were isolated in 20,502 M_2_ plants. The mutation frequency per unit dose was 2.03 × 10^−5^, while that induced by carbon ions in our previous study was 23.8 × 10^−5^ [47]. Shikazono isolated 88 tt (no purple pigmentation on leaves and stems) and gl (no trichomes on leaves) mutants from 104,088 M_2_ plants derived from carbon ions (150 Gy, 107 keV/μm), and the mutagenic effectiveness was estimated as being 5.64 × 10^−6^ [50]. However, when pollen was irradiated, the mutation frequency per dose was much higher than that using seeds. Nine mutants were isolated from 349 M_2_ individuals derived from carbon ions, while 35 mutants were isolated from 6321 M_2_ plants derived from gamma rays at 300 Gy, and the mutation effectiveness of the carbon ions and gamma rays was 17.2 × 10^−5^ and 1.85 × 10^−5^, respectively [20]. These results indicated that it might be more effective to use pollen, zygote after fertilization or the isolated single cells than that use the seeds as the starting material for mutagenesis. Since pollen is single-cell, and functions as a genetic information transmitter, the mutations in pollen could come directly to whole progeny, while treatment of the seeds usually induced chimeric mutations, and only few of them can transmitted to the next generation. A higher genome mutation frequency was observed in progeny derived from irradiation of reproductive organs than from seeds irradiation in pepper [38]. In other dicotyledonous plants, such as *Capsicum annuum*, the most prevalent mutation type induced by gamma rays (100 Gy) was dwarf mutants. The total mutation frequency of dwarfism and male sterility in M_2_ (1836 plants) was 7.09%, the mutation effectiveness was 70.9 × 10^−5^ per dose [13]. In the case of three *Lathyrus sativus* varieties that irradiated by 400 Gy gamma rays, the total phenotype mutation frequency was 4.89% (12.23 × 10^−5^ per dose) [16]. In monocotyledons plants, 740 and 1384 viable mutants were isolated in rice M_2_ population (90,386 plants) that derived from gamma rays (100–500 Gy) and electron beam (200–600 Gy) irradiated seeds, the mutation frequency was 2.35% [51]. In the case of maize, 10 salt-tolerant lines were isolated from 2248 M_2_ plants that derived from seeds irradiated by gamma rays at 100 Gy, the mutation frequency was 0.44% (4.44 × 10^−5^ per dose) [52]. In barley, one dwarf mutant was isolated from 10,000 M_2_ individuals that derived from seeds irradiated by 200 Gy gamma rays, the mutation frequency was 0.01% (0.05 × 10^−5^ per dose) [34]. In sorghum, the frequency of the chlorophyll mutations (albino, xantha, viridis, chlorina and variegated mutations) induced by 300 Gy gamma rays was 1.46% (4.86 × 10^−5^ per dose) in the M_2_ [19]. One semidwarf plant was isolated from 240 M_2_ plants that derived from seeds irradiated by 500 Gy of gamma rays in *Fagopyrum tataricum* Gaertn, the mutation effectiveness for semidwarf mutation was 0.83 × 10^−5^ per dose [53]. In *Brachypodium distachyon*, 31 lignin-deficient mutants were obtained in M_2_ populations (1668 plants) derived from seeds that acutely or chronically irradiated by gamma rays at doses of 50, 100, 150, 200, and 250 Gy, and the mutation frequency was 1.86% [22]. It was reported that the treatment of homozygous and heterozygous material affected the mutation frequency and spectrum. In the *Cymbidium* hybrids RB003 (heterozygosity), the mutation frequency of 50 Gy gamma rays was 0.35%, the mutation effectiveness is calculated as 7 × 10^−5^ per dose, which was higher than the homozygous materials [17].

### 3.2. Genomic Mutation Profile of Mutations Induced by Gamma Rays Irradiation

The molecular mutation profile at the genome scale can also be used for selecting a proper mutagen for mutation breeding and gene function analysis. Re-sequencing analysis of progeny of mutagenized plant material has provided a new perspective aimed at characterizing the mutation molecular profile of various physical mutagens at the whole-genome scale in the last 10 years [31,33,36,40,41,47,54,55,56,57,58,59,60,61]. In earlier research, only one mutant line was sequenced since it was still costly to perform NGS analysis. Therefore, thousands of mutations were output in a single plant at that time. The result was interfered by the original differences between the earlier released reference genome and the WT genome used for mutation treatment. Since Belfield reported the genome sequence of six mutants created using fast neutron in *Arabidopsis thaliana*, massively parallel sequencing of multiple samples derived from various types of irradiation has been used for comparative analysis [61]. Filter strategies work well to rule out the shared mutations between any two or more samples, followed by IGV or dideoxy sequencing of PCR products to verify the remaining mutations [62,63].

To date, there is no definite standard regarding which line and generation should be used for re-sequencing. Some studies have used randomly selected M_2_ individuals [31,40,64]; some have preferred higher-generation plants displaying visibly mutated phenotypes [35,41,55,58,65]. Few studies were performed in the M_1_ generation, as M_1_ plants are genetic chimeras. When exposed to mutagens, individual cells of the embryo are independently affected and divide to form cell populations with unique DNA damage or mutations. The white or light green color displayed in M_1_ leaves is one of the best examples of a genetic chimera induced by irradiation. As most plants propagate by seed, only mutations in generative cells can contribute to genetic variation in the non-irradiated offspring. The genomic mutations induced by physical mutagens that were output by NGS analysis mainly consist of SBSs and small InDels, while SVs have been reported sporadically. On the one hand, limited by the short insertion size of about 350 bp of library preparation and 2×150 bp short reads generated using NGS, and the positive rate for SV detection is still much lower than that for SBS and small InDel detection. For instance, Ichida detected only 3.0 inversions from 110 M_2_ rice lines derived from 150 Gy carbon-ion beam irradiation [64]. Only one inversion was detected in 19 M_2_ individuals of NHEJ-deficient *Arabidopsis thaliana* mutants (AtKu70^−/−^ and AtLig4^−/−^) that were irradiated with gamma rays (100 Gy) [60]. In rice, Li detected four SVs in seven M_5_ plants with visible mutated traits. On the other hand, SVs, especially large-scale rearrangements caused by severe damage induced by radiation, might reduce the transmission of gametes [20]. In addition, focusing on the mutations that can be transmitted to the next generation, we mainly analyzed the SBSs and small InDels since the M_3_ lines were used in the present study.

A total of 232 mutations, including 151 SBSs, 57 deletions, and 24 insertions, were detected in ten M_3_ progeny genomes. Among them, 98 were heterozygous and 134 were homozygous, which is in accordance with the fact that the theoretical number of homozygous mutations in the M_3_ generation should be 1.5 folds that of heterozygous mutations based on Mendel segregation genetic laws. On average 33.14 mutations in each individual line were caused by gamma rays at 1000 Gy. The mutation frequency per unit dose was 2.78 × 10^−10^ /bp/Gy. Shirasawa reported that 455 mutations (354 SBSs and 101 InDels) were detected in three tomato mutant lines derived from gamma rays at 300 Gy based on the Illumina HiSeq 1000 results, with an average of 151.7 mutations pre individual. Its mutagenic effectiveness was calculated as 5.32 × 10^−10^ /bp/Gy [10]. Yang detected 381 mutations (259 SBSs, 68 InDels and 54 multiple nucleotide variants) in four M_6_ rice mutants induced by gamma rays at 250 Gy based on the Illumina HiSeq 2500 results; on average 95.25 mutations were detected in each line and the mutation frequency was estimated to be 8.94 × 10^−10^ /bp/Gy [35]. Li detected, on average, 57.0 SBSs, 17.7 deletions, and 5.9 insertions in each M_5_ rice line derived from gamma rays (250 Gy) based on Illumina HiSeq X Ten results; the mutation frequency was reckoned as 9.11 × 10^−10^ /bp/Gy [41]. When compared with other physical mutagens, we prefer carbon ions (43 MeV/u, 50 keV/μm, 200 Gy) which under 69% relative survival rate, to gamma rays (relative survival rate, 62%) based on our previous study, since the mutation screening and detection methods were performed using the same standards. Carbon ions beam induced 444 mutations (320 SBSs and 124 InDels) in 11 M_3_
*Arabidopsis thaliana* individuals, in total, based on HiSeq 2500 platform; 40.3 mutations, on average, were detected in each line and the mutation frequency was calculated as 1.69 × 10^−9^ /bp/Gy (Du et al., 2017a). Based on Illumina HiSeq 2500 and 4000 (2 × 150 bp), Kazama detected 225 (146 SBSs, 79 InDels) and 455 (333 SBSs, 122 InDels) mutations in eight M_3_ pools derived from argon ions (290 keV/μm, 50 Gy) and carbon ions (30.0 keV/μm, 400 Gy), respectively, with each pools containing an average of 28.1 (4.70 × 10^−9^ /bp/Gy) and 56.9 (1.19 × 10^−9^ /bp/Gy) mutations, respectively [58]. Hase detected 228 (1.82 × 10^−9^ /bp/Gy) mutations (98 SBSs, 114 InDels and 16 complex mutations) in six M_2_
*Arabidopsis thaliana* plants using dry seeds irradiated by carbon ions (17.3 MeV/u, 107 keV/μm, 175 Gy) based on the Illumina NextSeq500 system [40]. In *Brachypodium distachyon*, 906, 1057, and 978 genome mutations (SBSs and 1–2 bp InDels) were detected in three dwarf and seed color mutants induced by gamma rays, respectively [37]. The mutagenic effectiveness of gamma rays at the genome scale was at 10^−10^, which is one order of magnitude lower than that of heavy-ion beam irradiation. Jo demonstrated that the progeny of irradiated gametophytes had similar or higher genome mutation frequencies than that of irradiated seeds in pepper by using genotype-by-sequencing [38]. All these studies indicated that the mutagenic effectiveness is dependent on multiple factors, including the radiation types, linear energy transfer (LET), dose, genome size, plant tissues, which generations used for re-sequencing analysis, even the NGS platform and the algorithms used for mutation calling. This characteristic reflects the flexibility of physical mutagens in mutation breeding and gene function studies. The magnitude of mutations can be regulated by adjusting the radiation parameters, and it can provide a balance between the demands for sufficient mutations and fast fixed agronomic traits.

### 3.3. Molecular Nature of Mutations Induced by Gamma Rays

It is widely accepted that ionizing irradiation can cause DSBs in genomic DNA, and NHEJ is the predominant pathway for DSBs repair. NHEJ can be classified into C-NHEJ and alternative NHEJ (A-NHEJ) pathways, both contributing to InDels or rearrangement mutations. In animals and plants, the DSBs repair pathways are affected by LET [66,67]. It was reported that the higher the LET, the less inclined the C-NHEJ pathway for DSBs repair in *Arabidopsis thaliana* [58,68]. Mutants deficient in Ku70/80 and Lig4, which are key members for C-NHEJ, exhibit higher root sensitivity to gamma-ray irradiation than the WT [6,61]. C-NHEJ-mediated rejoining events usually depend on 0 to 4 bp terminal microhomology or homology, while the A-NHEJ-mediated events depend on up to 25 bp [3]. C-NHEJ-deficient *Arabidopsis thaliana* mutants (AtKu70^−/−^ and AtLig4^−/−^) used increased size of terminal microhomology at the rejoining sites generated by gamma rays [60].

In the present study, the most frequent flanking sequence of the single-base InDels was homopolymeric (52.63%, 26/54), implying that these mutations might be highly related to the DNA replication slippage induced by irradiation. However, InDels with a size ≥2 bp were always accompanied by polynucleotide repeats (66.67%, 18/27) of 2 to 6 bp at their junctions, and the 2-bp microhomology seems to be more prevalent than the others (Figure 5D). Our present study does not contradict the characteristics of the flanking sequence of InDels observed in several other studies in *Arabidopsis thaliana*, and it is in accord with the basic characteristics and rules of C-NHEJ. Belfied’s study showed that 11 out of 22 InDels (≥2 bp) were induced by fast neutron located in a polynucleotide repeat context, while five of them were associated with the homopolymeric sequences [61]. In our previous study on carbon ions, we found the same trends [69]. Hase found that 63 out of 129 deletions larger than 2 bp induced by carbon ions were associated with polynucleotide repeats at the junction, while 14 out of 129 were involved in homopolymeric sequences. He also found that microhomology was less commonly involved in deletion mutations that were larger than 50 bp [40]. The microhomology at the rejoined sites of deletions peaked at 2 bp in the WT plants irradiated by gamma rays at 1000 Gy [60]. Although the concrete mechanisms remain unclear, our results suggest that the NHEJ is one of the main pathways that generate InDel mutations in gamma rays mutagenesis, but different mechanisms are used for the formation of single-base InDels and ≥2 bp InDels.

The transition to transversion ratio (Ts/Tv) in SBSs was 1.6, and the most frequent SBSs in the present study was G/C>A/T type (Figure 5A). In *Arabidopsis thaliana*, based only on the homozygous mutations, the ratio was 1.3 and 1.2 in the M_2_ lines derived from gamma rays at 1000 Gy and 1500 Gy, respectively [31]. The ratio value was 1.0, 0.89, 0.99, and 0.88 for the fast neutron (60 Gy), argon ions (290 keV/μm, 50Gy), and carbon ions (30.0 keV/μm, 400Gy; 50.0 keV/μm, 200Gy; 107 keV/μm, 175Gy) irradiation, respectively [40,58,61,65,69]. In rice, the Ts/Tv was 0.90 in carbon ions (135 MeV/u, 30 keV/μm), 2.4 for carbon ions (25.9 MeV/u, 76 keV/μm, 40 Gy), 1.4 for fast neutron (20 Gy), 1.49–1.6 for gamma rays (250Gy) [35,41,55,64,70]. In tomato, the ratio was 0.86 for gamma rays (300Gy). All of these values were much lower than the spontaneous (2.41) in *Arabidopsis thaliana* and the EMS-induced Ts/Tv ratios in tomato (2.60–2.94) [10,71,72]. G/C>A/T was the most frequent SBSs induced by EMS by promoting the alkylation of guanine, which resulted in a mispairing with thymine [73,74]. However, physical mutagens exhibit a much lower base bias of G/C>A/T than EMS. This indicates that different mechanisms are used for the induction of SBSs by physical and chemical mutagens. Irradiation can lead to the formation of cyclobutane pyrimidine dimer products at the pyrimidine dinucleotide locus. The C and 5-methyl-C within the dimer are unstable and easily deaminated to U or T, finally leading to C to T mutations [75]. In the present study, all the 28 C>T events proceeded to sequence context analysis, and 18 events (64.29%) exhibited the pyrimidine dinucleotide sequence. This is comparable to the proportions observed for the C>T mutations induced by fast neutron (70%, 14/20) and carbon ions (67.2%, 37/55) in *Arabidopsis thaliana*, and gamma rays (64.29%, 27/42) in rice [35,61,69]. On the other hand, hydroxyl radicals produced by irradiation can promote the generation of 8-hydroxy-2′-deoxyguanosine, which could pair with bases other than C, finally inducing a G/C>T/A mutation. However, the mechanism underlying the induction of other types of SBSs is still unclear. Here, we also examined the characteristics of the sequence context of all the SBSs. Interestingly, 33.11% (50/151) of the SBSs involved in terminal mirohomology. The non-G/C>T/A events showed a higher proportion of terminal microhomology than the G/C>T/A ones. However, we still could not conclude whether the terminal mircohomology is responsible for the none-G/C>T/A mutations, and further research is needed to understand the mechanism underlying these mutation types in the future.

### 3.4. Density of Affected Genes

In eggplants mutated by EMS, over 1620 genes had high-impact mutations. Although the high mutation gene densities achieved in the EMS-treated progeny ensure the recovery of allelic variants in most genes, this high mutational load may present a challenge for functional genomics studies, as well as for practical crop improvement. This is especially the case when trying to improve elite materials for a certain trait or a few traits since the presence of numerous undesirable mutations in the background may disrupt the finely tuned genetic architecture of the elite variety. In the present study, there were only nine individual genes, on average, that were predicted to have a higher chance of functional alteration in each line. Based on the results of ionizing radiation for mutation breeding, it was predicted that only a small number of genes would be functionally affected in the genomes, as a trait of interest can be changed without the other traits being impacted, resulting in a relatively short breeding cycle. Only five to 11 genes were detected to have homozygous mutations in three argon-ion-induced *Arabidopsis thaliana* mutant lines [65]. In our previous study, three to 15 genes were found to be affected by carbon ions in the M_3_ lines [69].

## 4. Conclusions

In conclusion, the frequency and spectrum of gamma ray-induced mutations in *Arabidopsis thaliana* that were revealed based on phenotype screening and whole-genome re-sequencing analysis of the phenotype-selected progeny were provided in the present study (Figure 7). A phenotypic mutation collection was constructed, which included 417 (2.03%) plants with visible mutated phenotypes, and the phenotypic mutation effectiveness of gamma rays was 2.03 × 10^−5^ per dose in *Arabidopsis thaliana*. However, in the present study, the mutants that would be visible under exogenous stimuli, such as salt, drought, ultraviolet light, pathogenic bacteria, and so on, were not taken into account. Therefore, the actual phenotypic mutation rate and spectrum might be higher than the results presented here. In each individual genome, there were 21.6 SBSs, 11.6 small InDels, and the function of nine protein-coding genes, on average, were predicted to be affected by sequence variation. The genomic mutation frequency was 2.78 × 10^−10^/bp/Gy in the M_3_ generation. The ratio of Ts/Tv was 1.60, and single-base InDels accounted for 66.7% of the small InDels. Examination of the flanking sequence at the mutations revealed that the SBSs and small InDels had distinct characteristics at their rejoint sites (i.e., the frequency of pyrimidine dinucleotide and terminal microhomology and its size). This suggests that different mechanisms are responsible for SBSs and InDels. Mutation categories are affected by multiple factors, such as the parameters of physical mutagens, as well as the plant species. Comparison with previously reported studies on the mutagenic effects of different physical mutagens, different plant species, and the common and specific nature of the mutation profile was also present here. This characteristic reflects the flexibility of physical mutagens in mutation breeding and gene function studies. These results will aid in the full understanding of the mutagenic effects and mechanisms of gamma rays and provide a basis for suitable parameter design for mutation screening, which can further facilitate the development of more controlled mutagenesis methods for plant mutation breeding.

## 5. Materials and Methods

### 5.1. Plant Material and Growth Condition

*Arabidopsis thaliana* of Col-0 ecotype type were used in this study. Plants were grown in the culture room with the condition of 22 ± 2 °C, 60% relative humidity under illumination of 108 μmol/m^2^/s at an 18 h light/6 h dark photoperiod.

### 5.2. Gamma Rays Irradiation, Plant Growth, Biological Effects on M_1_ Plants

Dry seeds were placed into the centrifuge tubes regarding the strong penetration of gamma rays, the samples were irradiated by gamma rays (cobalt-60) at doses of 0, 100, 200, 300, 400, 500, 600, 800, 900, 1000, 1100, 1200, 1300, 1400, 1500, and 1600 Gy with dose rate of 8.5 Gy/min at the Beijing Radiation Center.

### 5.3. Biological Effects on M_1_ Plants

The survival rate and DNA polymorphic rate (by using ISSR and RAPD techniques) were determined and shown in our previous study [12]; here, the survival rate was presented as a value relative to that of the control group. About 50 seeds were surface sterilized in 20% bleach for 10 to 12 minutes, washed five times with sterile water and suspended in sterile 0.1% agarose, then sown on Murashige and Skoog medium (0.8% agar and 1.5% sucrose, sterilized at 115 °C for 30 minutes) at room temperature. After vernalization in the dark for 72 h at 4 °C, the plates were transferred to a culture room (22 ± 2 °C) and were kept vertically oriented placement. For primary root length assay, the seeds were cultured with normal illumination of 108 μmol/m^2^/s at an 18 h light/6 h dark photoperiod, while for hypocotyl length assay, the seeds were cultured in darkness. Seven days later, the seedlings were photographed using a digital camera, the length of roots and hypocotyls were measured using Image J 1.46 software.

### 5.4. Mutation Screening

In total, 2070 M_1_ seeds irradiated by gamma rays of 1000 Gy were sown, 1139 M_2_ families were harvested from M_1_ individuals, and 18 seeds from each M_2_ family were sowed as the mutation screening population. The plants that displayed different visible phenotypes from that of WT were recorded as candidate mutant lines and given a unique number. Ambiguous candidate mutant lines were sown to verify whether these traits could be inherited by their progeny. Based on this, the M_2_ mutation type and rate were corrected.

### 5.5. Whole-Genome Re-Sequencing and Mutation Analysis

In total, 2070 M_1_ seeds irradiated by gamma rays of 1000 Gy were sown, 1139 M_2_ families were used for whole-genome re-sequencing. Genomic DNA of a single plant was extracted using a general cetyltrimethylammonium bromide (CTAB) protocol. Re-sequencing was performed by using an Illumina HiSeq 4000 platform at Biomarker Technologies Company (Beijing, China). Low-quality reads with an adaptor sequence were removed, and then the clean data were used for further bioinformatics analysis. In short, the clean data were mapped to the reference genome (TAIR10) using BWA [76]. SBSs and small insertions and deletions (small InDels) were called by the combined use of SAMtools and VarScan 2 [77,78]. Qualimap 2 was used to evaluate the alignment data [79]. The basic information of the alignment quality is shown in Appendix A. The standards used for homozygous and heterozygous mutation, real mutations, and false mutations were following the same ways as described previously [69]: 1) Variants shared between two or more lines were filtered out as differences already existing between the lines we originally used for irradiation and the reference genome; 2) The variants of allele frequency (AF) ranging from 25% to 75% were called as heterozygous mutations, and those with an AF ≥ 75% were regarded as homozygous mutations; 3) The reads supporting the variant call whose number was below three were filtered; 4) After the above three steps, all the output mutations were verified using IGV (Appendix A) [80]. The positive mutations were annotated using SnpEff v.4.2 [81]. KEGG pathway and Gene Ontology (GO) analysis of affected genes were performed using KOBAS 3.0 and DAVID Bioinformatics Resources 6.8 [82,83].

## Figures and Tables

**Figure 1 ijms-23-00654-f001:**
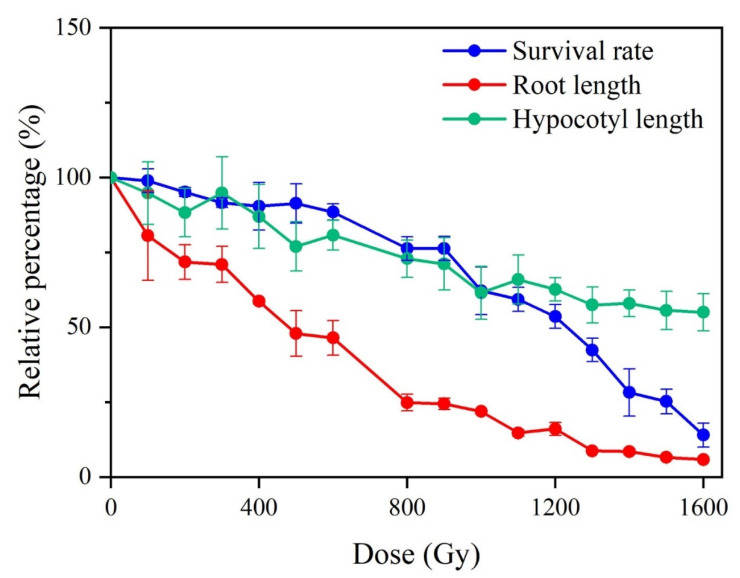
Biological effect of gamma rays irradiation on M_1_ plants. All the data indicated the relative value to the control group. Each data point was mean ± standard error.

**Figure 2 ijms-23-00654-f002:**
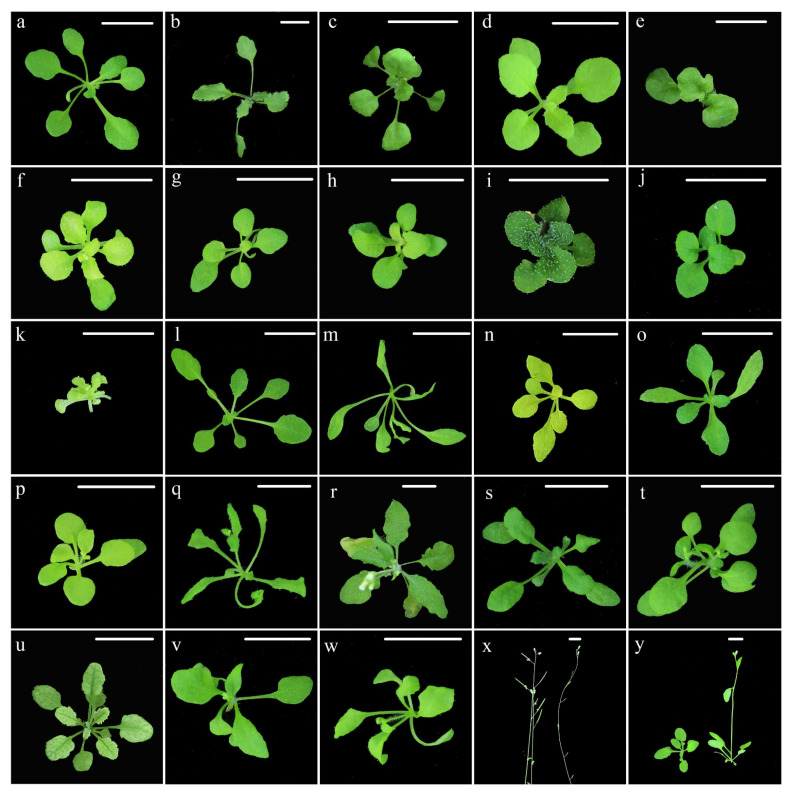
Phenotype of partial mutants induced by gamma ray irradiation. (**a**), Wild type; (**b**–**y**), mutants. For (**x**,**y**), the left plant is WT, the right plant is the mutants. Scale bars, 10 mm.

**Figure 3 ijms-23-00654-f003:**
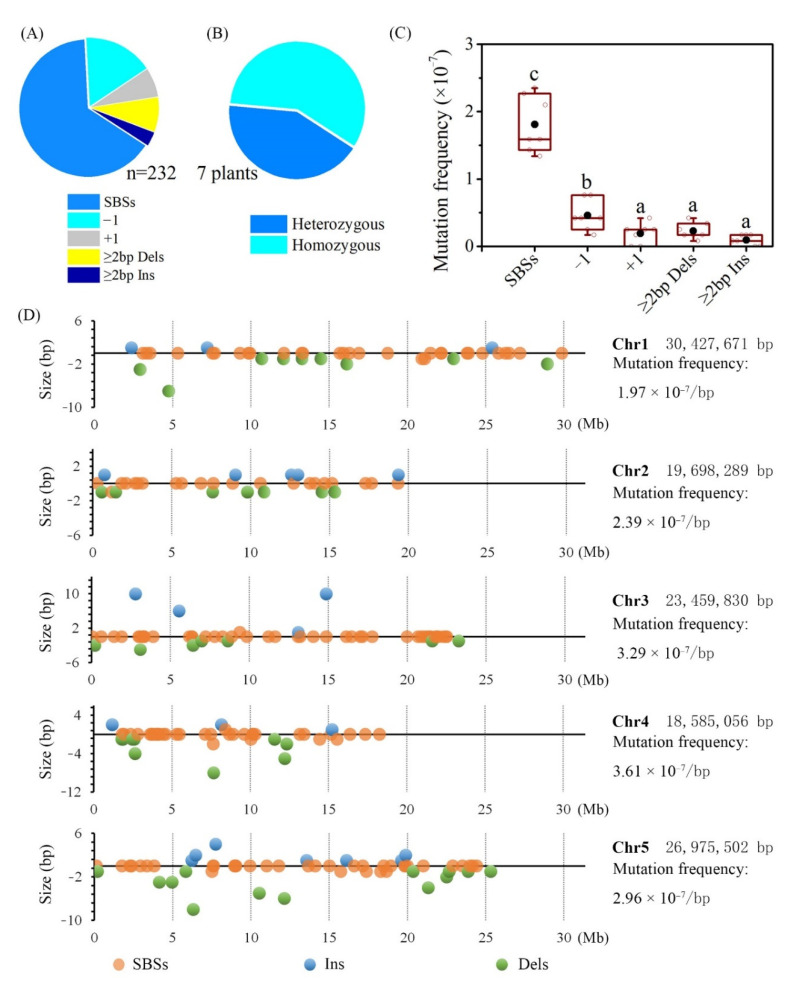
Summary of identified DNA sequence mutations. (**A**), Mutation type; (**B**), zygosity of mutations; (**C**) mutation frequency in M_3_ genome, data followed by the same alphabetic letters are not significantly different between any two of the mutation types (*p* > 0.05) by Duncan’s multiple range test; (**D**), the distribution of mutations on chromosomes.

**Figure 4 ijms-23-00654-f004:**
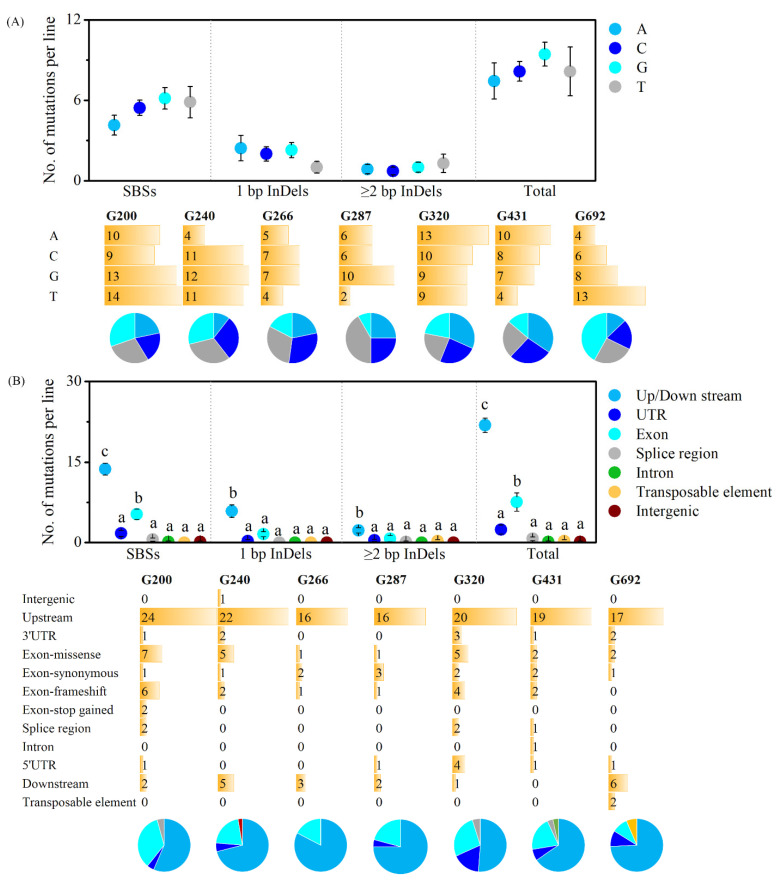
Nucleotide bias of mutations (**A**) and distribution of mutations on gene structure (**B**). Data of scatter plot are mean ± standard error from seven replicates. Data followed by the same alphabetic letters are not significantly different between any two of the mutation types (*p* > 0.05) by Duncan’s multiple range test.

**Figure 5 ijms-23-00654-f005:**
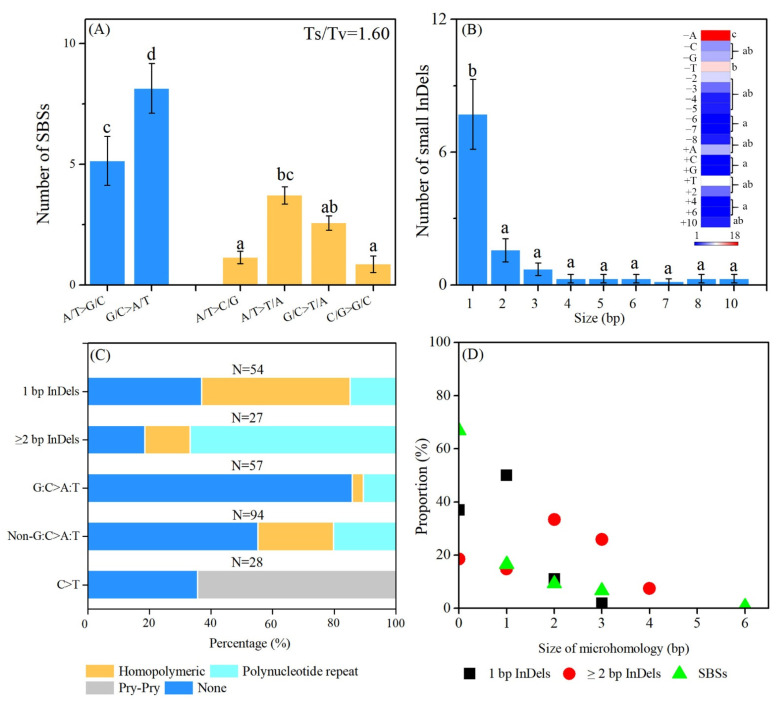
Characteristics of SBSs and small InDels induced by gamma rays. (**A**), Categories of SBSs. (**B**), size distribution of InDels. Each data point was mean ± standard error from seven replicates. Data followed by the same alphabetic letters are not significantly different between any two of the mutation types ((*p* > 0.05) by Duncan’s multiple range test. (**C**), Characteristics of preferential sequences flanking the DNA mutations. (**D**), Distribution of the size of microhomology observed at Indels and SBSs. Details of flanking sequences are shown in Appendix A.

**Figure 6 ijms-23-00654-f006:**
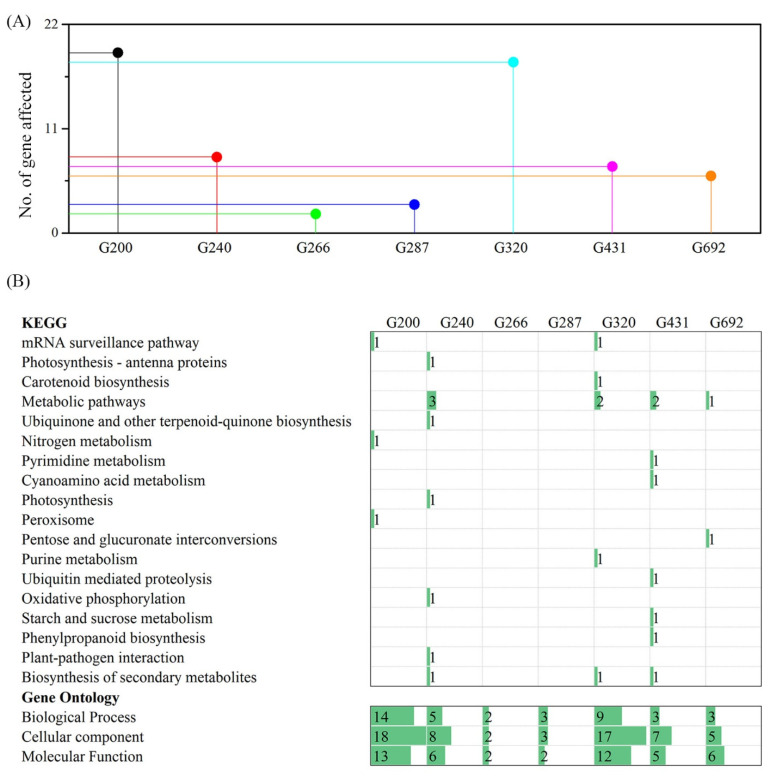
Genes affected by gamma rays irradiation in *Arabidopsis thaliana*. (**A**), Number of genes affected in each line. (**B**) KEGG pathway and GO analysis of affected genes.

**Figure 7 ijms-23-00654-f007:**
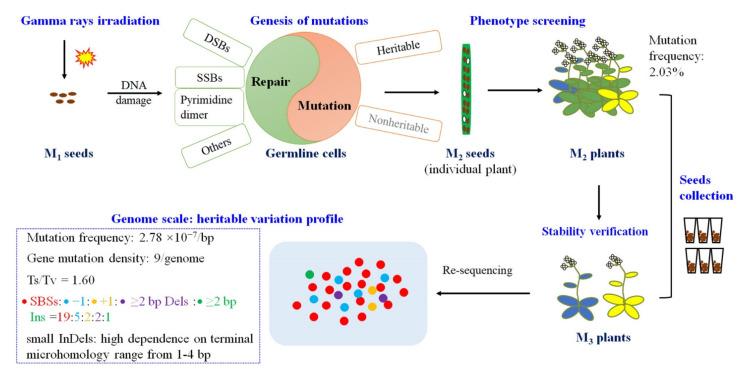
The mutation profile of mutations induced by gamma rays in *Arabidopsis thaliana*.

**Table 1 ijms-23-00654-t001:** Classification of visible mutation phenotype induced by gamma rays in M_2_ generation.

Category	No. of Mutant	Frequency (%)	Brief Description
Leaf	267	1.30	
Lamina	134	0.65	The ratio of length to width, the edge and margin, surface of lamina;
Arrangement	12	0.06	Non-radial symmetry arranged rosette leaves, number of leaves
Petiole	45	0.22	Changes in the length of petioles
Vein	13	0.06	Obvious vein
Color	63	0.31	Yellow, variegated or albino leaves
Seeds viability	36	0.18	All the 18 seeds from single M_2_ family did not germinate
Stem	29	0.14	Dwarf
Fertility	10	0.05	Short or small siliques
Premature	75	0.37	Early bolting or flowering between 5–10 days than the Lab-WT
Total	417	2.03	

**Table 2 ijms-23-00654-t002:** Affected genes with non-synonymous ^a^, UTR and splice region mutations the re-sequenced lines.

Line	ID	Description
G200	AT1G10170	NF-X-like 1 (NFXL1)
	AT1G13870	KTI12-like, chromatin associated protein (DRL1)
	AT1G50600	Scarecrow-like 5 (SCL5)
	AT2G17480	Seven transmembrane MLO family protein (MLO8)
	AT2G33200	F-box family protein
	AT3G02810	Protein kinase superfamily protein
	AT3G05110	Early endosome antigen-like protein, putative (DUF3444)
	AT3G09100	mRNA capping enzyme family protein
	AT3G09750	Galactose oxidase/kelch repeat superfamily protein
	AT3G09960	Calcineurin-like metallo-phosphoesterase superfamily protein
	AT3G45060	High affinity nitrate transporter 2.6 (NRT2.6)
	AT3G48140	B12D protein
	AT3G57430	Tetratricopeptide repeat (TPR)-like superfamily protein (OTP84)
	AT4G13190	Protein kinase superfamily protein
	AT4G15200	Formin 3 (FH3)
	AT4G23350	Transmembrane protein, putative (DUF239)
	AT5G10620	Methyltransferase
	AT5G25840	DUF1677 family protein (DUF1677)
	AT5G43280	Delta (3,5), delta (2,4)-dienoyl-CoA isomerase 1 (DCI1)
G240	AT1G10720	BSD domain-containing protein
	AT2G30520	Phototropic-responsive NPH3 family protein (RPT2)
	AT2G32410	AXR1-like protein (AXL)
	AT3G59350	Protein kinase superfamily protein
	AT3G60700	Hypothetical protein (DUF1163)
	AT4G04640	ATPase, F1 complex, gamma subunit protein
	AT2G34420	Photosystem II light harvesting complex protein B1B2 (LHB1B2)
	AT4G23660	Polyprenyltransferase 1 (PPT1)
G266	AT1G60090	Beta glucosidase 4 (BGLU4)
	AT5G63100	S-adenosyl-L-methionine-dependent methyltransferases superfamily protein
G287	AT3G01530	Myb domain protein 57 (MYB57)
	AT4G17140	Pleckstrin homology (PH) domain-containing protein
	AT4G18110	RING/U-box superfamily protein
G320	AT1G56360	Purple acid phosphatase 6 (PAP6)
	AT1G77020	DNAJ heat shock N-terminal domain-containing protein
	AT2G01660	Plasmodesmata-located protein 6 (PDLP6)
	AT2G29790	Maternally expressed family protein
	AT3G20660	Organic cation/carnitine transporter4 (OCT4)
	AT3G25727	Non-LTR retrolelement reverse transcriptase
	AT3G56320	PAP/OAS1 substrate-binding domain superfamily
	AT3G58300	Phospholipase-like protein (PEARLI 4) family protein
	AT4G04880	Adenosine/AMP deaminase family protein
	AT4G08350	Global transcription factor group A2 (GTA2)
	AT4G26650	RNA-binding (RRM/RBD/RNP motifs) family protein
	AT4G39190	Nucleolar-like protein
	AT5G05820	Nucleotide-sugar transporter family protein
	AT5G41460	Transferring glycosyl group transferase (DUF604)
	AT5G50070	Plant invertase/pectin methylesterase inhibitor superfamily protein
	AT5G51730	RNA-binding (RRM/RBD/RNP motifs) family protein
	AT5G52570	Beta-carotene hydroxylase 2 (BETA-OHASE 2)
	AT5G55970	RING/U-box superfamily protein
G431	AT1G15490	Alpha/beta-Hydrolases superfamily protein
	AT1G70320	Ubiquitin-protein ligase 2 (UPL2)
	AT3G18080	B-S glucosidase 44 (BGLU44)
	AT3G18680	Amino acid kinase family protein
	AT4G08580	Microfibrillar-associated protein-like protein
	AT5G22580	Stress responsive A/B Barrel Domain-containing protein
	AT5G22794	Hypothetical protein
G692	AT1G09040	Arginine-glutamic acid dipeptide repeat protein
	AT1G44750	Purine permease 11 (PUP11)
	AT2G15730	P-loop containing nucleoside triphosphate hydrolases superfamily protein
	AT3G10015	tRNA
	AT3G10720	Plant invertase/pectin methylesterase inhibitor superfamily
	AT3G57910	D111/G-patch domain-containing protein

^a^ missense, frameshift, stop gained and stop lost were included.

## Data Availability

The genome-wide sequencing data reported in this research have been deposited in the Genome Sequence Archive in BIG Data Center, Beijing Institute of Genomics (BIG), Chinese Academy of Sciences, under accession number CRA000585 that was publicly accessible in the website (http://bigd.big.ac.cn/gsub/index.jsp (accessed on 6 November 2019)).

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
