# Peer review of "Frequency and Spectrum of Mutations Induced by Gamma Rays Revealed by Phenotype Screening and Whole-Genome Re-Sequencing in Arabidopsis thaliana"

_ijms, 2022, doi:10.3390/ijms23020654_

Round 1

Reviewer 1 Report

I am impressed with the contribution the authors put into this research. However, I miss a broader reference to plant models other than Arabidopsis in case frequency and spectrum of gamma ray-induced mutations. I noticed only e.g. Cymbidium,  Oryza sativa and Solanum lycopersicum. If the authors in the discussion will able to show differences between other models among monocotyledons (especially please compare to rice studies) and dicotyledons, this would make this work more attractive.

Reviewer 2 Report

This is a very interetsing research. Authors describe very precisely analysis of the effect of physical mutagenesis on the type of mutation, frequency, nature of change in DNA structure.

Comments:

 Line 15: it is better to write diversity, not innovation.

Line 20:“biological effect“?

Line 30-31: „Nine individual genes, on average, that were predicted to have a higher chance of functional alteration“. Please, re-formulate.

Line 42: „high dependence on ornamental plants“ ?

Line 133: „The lethal effect“ can not be determined,  your determined seedling „viability“.

Line 135: „root growth was the most sensitive growth indicator“ ? Please, formulate better!

„the elongation of the primary root was completely inhibited“ – growth can be inhibited, not elongation.

Line 116: „Many mutants were isolated during mutagenic effect research“ – please.re-formulate

It will be nice to mention also some other methods commonly used for increase genetic diversity like CRISPR..

Line 145: „primary length“  = root length.

Figure 1: please, on the panel A explain better: that you mean root length and hypocotyl length.

Panel B seems to be a direct copy from previous paper, this is not possible to show it again, just cited!

Line 181: it seem to be repetition.

It is better to re-arrange figure 2 accordingly to order of the lines your described.

Lines 291-296: this is a common well-known points, can be avoided,

Line 297: what do you mean as „embryos were damaged by gamma rays“?

Root growth requires de novo cell divsion, what is defenitely more sensitive to any mutagens, but hypocotyl elongation is only cell elongation. You can mention this point here.

Moreover, you can alos rescue plants with defcet in root system by addition of sucrose and generation seeds in vitro (https://www.biorxiv.org/content/10.1101/2020.08.23.263491v1.full).

Line 305: „starting material mutagens“?

You can also add points that pollen is a single cell and mutation come directly to whole progeny, while treatment of the seeds induced „mosaic“ mutation, and only few of them can come to the next generation.  The most effective way of mutagensis, of course, is single cell mutation like pollen, zygote after feritilization or isolated single cells like protoplasts.  

Line 316: The mutation frequency of the chlorophyll mutations ??

Line 325: „mutagen for mutation breeding“??

Line 329: „thousands of mutations were produced in single plant at that time“ ??

Line 482: „comprehensive frequency and spectrum of..“???

„phenotype of the phenotype-selected“ ??

It will be also nice to mention that many „hidden“ mutation did not taking into account, including some what become visible under stress, or other treatmemts. So, one you described can be only the top of the iceberg…

Line 521: have you used full MS or 1/2MS? With sucrose or without sucrose? Please, convert lux to µmol light intensity.

Line 531: do you mean phenotype, not genotype?  

Round 2

Reviewer 2 Report

The text is better now, some minor corrections still need.

Line 18: „mutation breeding by gamma rays“ ? in addition, you repeat twice gamma rays in one sentence,

Line 46: I am not sure about last century, it is better to write more precise, like last decades…

Line 62: „Lpa5, Lpa9, and Lpa59“ – is it mutants? Please, mention this point.

Line 388: „ mutation frequency of the chlorophyll mutations“ – I mean you can keep only „ chlorophyll mutations” but do not write “mutation.. of the … mutation”!

Line 609- 611: I think it will be more effective to use seeds with active chromatin (after germination), not dry seeds.

Line 619: please, add temperature.

Figure 1: I think it should be hypocotyl, not hypocotyle.
